# Active Components, Antioxidant, Inhibition on Metabolic Syndrome Related Enzymes, and Monthly Variations in Mature Leaf Hawk Tea

**DOI:** 10.3390/molecules24040657

**Published:** 2019-02-13

**Authors:** Zhuo Chen, Dan Zhang, Jia-Jia Guo, Wei Tao, Rui-Xue Gong, Ling Yao, Xing-Long Zhang, Wei-Guo Cao

**Affiliations:** 1College of Traditional Chinese Medicine, Chongqing Medical University, Chongqing 400016, China; 15095835920@163.com (Z.C.); zhangdan01234567@sina.com (D.Z.); gjj199509@163.com (J.-J.G.); Twei91001@163.com (W.T.); grx_cq@163.com (R.-X.G.); michealle_10@163.com (L.Y.); 2Center for Disease Control and Prevention of YuZhong District of Chongqing, Chongqing 400010, China; 18883937118@163.com; 3The Lab of Traditional Chinese Medicine, Chongqing Medical University, Chongqing 400016, China

**Keywords:** Hawk tea, active components, antioxidant, α-glucosidase and lipase, monthly variation

## Abstract

Hawk tea is a rich and edible resource, traditionally used as a beverage in South China. This drink has many pharmacologic effects, such as acting as an antioxidant and reducing blood sugar and lipids. The objective of this work was to explore the active compound contents, bioactivities and their monthly changes, and optimize the harvest time. In the present study, Hawk tea from each month in 2017 was collected and extracted with 70% (*v/v*) ethanol. The contents of the total flavonoids and total phenols were determined using the colorimetric method. We determined the contents of seven characteristic active substances—hyperin, isoquercitrin, trifolin, quercitrin, astragalin, quercetin, and kaempferol—using high-performance liquid chromatography. The crude extract was tested for its antioxidant and inhibitory properties on enzymes involved in metabolic syndrome. Specifically, 2,2-diphenyl-1-picrylhydrazyl, 2,2′-azino-bis (3-ethylbenzothiazoline-6-sulphonic acid), ferric-reducing power assay, and the inhibition capacity test on α-glucosidase and lipase were conducted to determine the antioxidant effect in vitro, as well as the reduction of blood sugar and lipids. Monthly variations in activities and components were determined by numeric analysis and comparison. Correlation analysis revealed that antioxidant effects are significantly correlated with the total flavonoids. The hierarchical cluster analysis of bioactivities and their contents indicates that October and November are the best harvesting months, which differs with the habitual collection of Hawk tea.

## 1. Introduction

*Litsea coreana* Levl. var. lanuginose, which used to be a wild plant, is known as Hawk tea. Hawk tea has become increasingly cultivated in the rural areas of South China, especially in Sichuan, Chongqing, and Guizhou. As one of the non-*Camellia* (Theaceae) teas (or in Chinese: Bie-yang-cha) [1,2], Hawk tea is caffeine (1,3,5-trimethylxanthine)-free [3], and its consumption now even surpasses that of green tea.

Hawk tea’s wide use dates back to classical China times due to its absence of bitterness. It possesses a mellow, sweet taste as a beverage, and is useful for disease prevention and medical treatment [4]. Hawk tea is an effective antioxidant and is used for the reduction of blood sugar and blood lipids, detoxification and detumescence, and improvement of eyesight and liver lipid metabolism [5]. This herb tea is divided into bud tea, primary, and mature leaf tea in terms of different maturity degrees, among which the latter is the cheapest form that accounts for the majority of production, hence there is interest in the mature leaves for health care and financial reasons.

Exogenous plant antioxidants have been a persistently topic of interest as massive amounts of free radicals result in chronic or life-threatening diseases [6]. Antioxidants can significantly delay or prevent the oxidation of oxidizable substrates [7]. Supplementation with exogenous antioxidants is one of the most promising methods for countering undesirable oxidative stress [8]. As a medicinal and edible resource, the water extract (for the polysaccharides) [9,10] of mature leaf Hawk tea and the aromatic constituents [5,11] have been verified to possess good antioxidant potential, and the fraction extracted by ethanol was proven in our previous work to exhibit strong antioxidant activities against 2,2′-azino-bis(3-ethylbenzothiazoline-6-sulphonic acid) (ABTS) radical, as well as 2,2-diphenyl-1-picrylhydrazyl (DPPH) radical [12].

Metabolic syndrome (MetS) is a state of insulin resistance, oxidative stress, and chronic inflammation that affects 25% of the world population [13]. Other than therapeutic lifestyle changes, pharmacological treatments and protection are important for remedying metabolic syndrome-related factors. The reduction of postprandial hyperglycemia by the inhibition of enzymes involved in carbohydrate metabolism (α-glucosidase and α-amylase), inhibition of lipolytic enzymes including lipase, inhibition of oxidative stress, and the delay of inflammatory process are the most common therapeutic approaches to treat metabolic syndrome [14]. Based on its folk usage for reducing blood sugar and blood lipids, Hawk tea is a potential source for treatment of MetS. For carbohydrate metabolism, extracts in different solvents were shown to effectively inhibit the activity of α-glucosidase [11,15,16], but few studies have comprehensively examined the effects both on glycolytic and lipolytic enzymes. 

The harvest season of Hawk tea is not fixed, due to the size of the producing regions. The process depends on the appearance, taste and regional collection habitats. Considering the pharmacological activities of this traditional drink, aside from factors such as the specific geographic area and postharvest handling method, collection time equally affects the quality [4]. Flavonoids are secondary plant metabolites considered to be the dominant phenolic phytochemical components according to research that elucidated the structures of the chemical constituents in Hawk tea [17,18,19]. Quercetin-3-D-galactoside and quercetin-3-rhamnoside, have been identified as the main flavonoid components [12,20,21,22]. For Hawk tea, researchers have attempted to distinguish the production region [15], the botanical origin [12], and the maturation degree [23] on the basis of the ingredients or activities. Even seasonal high-performance liquid chromatography (HPLC) fingerprint analysis has been conducted [24], but the amounts of chemicals and activity differences based on collection time have not been studied.

The objective of the present work was to investigate the quantities of active phenolic compounds, their antioxidant effect and inhibition of enzymes involved in metabolic syndrome and their monthly change features, and determine the optimum picking time based on scientific evidence. We wanted to provide a basis for the full use of the mature leaves. The leaf tea samples in the present study were harvested from the same geographic location and exposed to the same handling methods, but the picking month varied from January to December. Aside from the quantitative analysis of total phenols, total flavonoids, and seven monomeric flavonoids, in vitro antioxidant assays (ABTS, DPPH along with reducing power tests) were conducted to evaluate the antioxidant properties. The inhibition effects on the typical enzymes relevant to metabolic syndromes, such as α-glucosidase and pancreatic lipase, were assessed. As chemometric method, hierarchical cluster analysis (HCA) was used to classify the similar months based on active component contents and bioactivities. The aim of this study was to improve the use of Hawk tea and bolster its development, ultimately increase the economic viability of mountainous regions.

## 2. Results

### 2.1. Quantification and Comparative Analysis

#### 2.1.1. Total Flavonoid Content (TFC) and Phenolic Content (TPC)

Phenolic compounds are widely distributed in plants; the yields of the samples picked from different months varied widely. The TFC and TPC quantities were calculated from the regression equation of rutin (y = 0.9314x + 0.0397, R > 0.9997) and gallic acid (y = 0.011x + 0.1225, R > 0.9997). As indicated in Table 1, the November ethanol extract of Hawk tea had the highest flavonoid content at 11.75 ± 0.65 mg RE/g DW (mg rutin equivalents per gram of dry tea sample), and the TPC of the sample from August was 21.92 ± 1.87 mg GAE/g DW (mg gallic acid equivalents per gram of dry tea sample). The TFC and TPC for December were 6.50 ± 0.48 mg RE/g DW and 10.91 ± 0.5 mg RE/g DW, respectively, demonstrating inferiority in terms of that month’s polyphenol content. As the dominant component, TFC varied with an irregular trend. From the beginning of the year, TFC increased each month until March and April, then decreased for May and June. In July, TFC increased again to a higher value, but declined to a low level in the following month. TFC continued increasing from September to November and decreased again in December. The TFCs of the different months ranked as follows: Oct and Nov > Sep > Mar and Apr > Jul > Jan and Feb > May, Jun, Aug, and Dec. The TPCs were ranked as follows: Aug > Nov and Sep > Apr, July, and Oct > Feb and Mar > Jan, May, and Jun > Dec. Both the TFC and TPC from September to November showed a relatively increasing trend compared with the first half of the year, where March and April showed preferable results in terms of the total phenolic phytochemicals. 

#### 2.1.2. Flavonoid Components and Contents

The flavonoid components of all 12 samples were separated using the HPLC-photodiode array detector (DAD) method. For this operation, a chromatogram of November’s tea (Figure 1) was obtained at 350 nm. By comparing the retention times with reference substances, six peaks were qualitatively analyzed and quantified. The retention time of hyperin was 24.39 min, isoquercitrin was at 25.69 min, trifolin was at 29.91 min, and quercitrin and astragalin were both observed at 32.91 min. With their weak polarity, quercetin and kaempferol were observed at 49.72 min and 58.13 min, respectively. What distinguished the picking months was not the constituents but rather the individual and total percentages. For the total monomers contents, December and September were the richest months. The comparative contents of characteristic flavonoid compounds are listed in Table 2.

Both quercitrin and astragalin have been found to be important as antioxidants. Truong et al. [25] reported that quercitrin can act against acetaminophen (APAP)-induced hepatotoxicity through the enhancement of the activity and expression of antioxidant enzymes, and Cho et al. [26] reported that astragalin may be a potent agent in antagonizing endotoxin-induced oxidative stress, which can lead to airway dysfunction and inflammation. Among the detected components a rich amount of quercitrin and astragalin was uncovered, but the two monomers were superimposed in a much higher peak, even though a range of conditions were tested, which is inconsistent with previous qualitative analysis [15]. The difficulty in isolating the two isomers may result from the number of hydroxyls and similar polarity. Therefore, the two similar compounds were quantitatively analyzed by calculating their sum according to the approximate absorption coefficient values (350 nm). The total content of the two constituents was the highest throughout the entire year. The contents of the two components in September and November tea were the highest at 461.9 and 447.32 μg/g, respectively, followed by February (401.43), October (379.85), June (369.04), August (354.27), July (346.20), April (311.28), May (301.28), December (278.99), March (269.80), and January (232.63).

Among the remaining individual flavonoids, the structures of hyperin and isoquercitrin are similar, and their retention times are close to each other. Hawk tea from November (49.11 μg/g) had the most hyperin, followed by July (43.66 μg/g), whereas that in February was 28.59 μg/g. Isoquercitrin from September tea (137.35 μg/g) was quantified to be the most abundant, followed by November, at 122.99 μg/g. As for kaempferol-3-*O*-galactoside, also named trifolin, its contents ranged from 14.82 to 45.43 μg/g, belonging to April and November. Quercetin and kaempferol in the 12 samples showed lower concentrations, whereas the sample from November contained the largest amount of both flavonoids (11.02 and 26.57 μg/g, respectively), but the contents were still less than the other compounds in this sample. Notably, the total contents of the seven individual flavonoids did not show much difference, apart from the fact the November yield was about twice that of January.

### 2.2. Method Validation

Recovery, repeatability and the stability (0, 2, 4, 8, 12 and 24 h) tests were completed and well validated. The relative standard deviation (RSD) of recovery ranged from 97.54% to 100.97%, and the repeatability (retention time, RT; and retention peak area, RPA) was 0 to 1.13% among the six peaks. The RSD of the precision was no more than 1.00%, and the stability was less than 1.80%. The limits of detection (LOD) and limits of quantification (LOQ) of the six analytes, and the calibration curves are shown below in Table 3.

### 2.3. Antioxidant Activity Evaluation

#### 2.3.1. Scavenging Effect on DPPH and ABTS Radicals

DPPH is a stable free radical that is widely used to screen for antioxidant active components in fruits, vegetables and natural treatments [27]. The results in Table 4 show that when the half maximal scavenging concentration (IC_50_) of butylated hydroxytoluene (BHT) is 0.04 ± 0.001 mg/mL, mature leaves from January to December can scavenge the DPPH radicals with IC_50_ values from 0.14 ± 0.001 to 0.21 ± 0.001 mg/mL. The highest inhibitory activities among the 12 months were found in the concentration of 0.14 ± 0.01 from December, 0.15 mg/mL from October and April, and followed by 0.16 from March, July and September. We observed that the teas from January, May, June, August, and December contained relatively lower TFC, but the IC_50_ values were around the highest of all 12 months, revealing the variation in scavenging activity. The sample picked in January exhibited the weakest suppression ability.

The ABTS assay displayed a similar trend as the DPPH one. As shown in Table 4, the IC_50_ values obtained in October and November show a relatively stronger antioxidant ability, at 1.92 ± 0.02 and 2.05 ± 0.03 mg/mL, respectively, whereas the positive control BHT scavenged 50% of the ABTS at a concentration of 0.60 ± 0.01 mg/mL. In line with the result of DPPH-inhibiting assay, Hawk tea from August is a weaker antioxidant, with an IC_50_ value of 7.51 ± 0.29 mg/mL for the ABTS experiment, followed by that of May at 4.69 ± 0.03 mg/mL and December at 4.52 ± 0.18mg/mL.

#### 2.3.2. Radical Reducing Power Test

The reducing power of a bioactive substance is closely related to its antioxidant ability and is an effective indicator of potential antioxidant performance [28,29]. A higher absorbance indicates a stronger reducing power. Similarly, the trend in the reduction potential of the sample solution is that the optical density (OD) increases as the dose increases (from 0.5 to 10 mg/mL). 

The absorbances of 5 mg/mL solutions of each sample were illustrated to prove a clear contrast after the chromogenic reaction. When the OD of control BHT was 1.109 ± 0.042 AU, as the absorbance variation in Figure 2 indicates, leaf tea harvested in November (0.687 ± 0.005 AU) possessed the best reducing capability, but the result of August at 0.470 ± 0.019 AU revealed a relatively poor reducing effect. Overall, tea from May, August and December with the lowest absorbances, appear to be the worst months in terms of ferric reducing activity. For all three antioxidant assays, the scavenging activities changed in a similar direction, generally aligned with TFC. This finding hints at the good correlation between antioxidant properties and TFC.

### 2.4. Inhibition of Enzymes Involved in Metabolic Syndrome

#### 2.4.1. α-Glucosidase Inhibition

To take advantage of herbal materials, we tested the inhibition activity of a series of concentrations, and PBS (without tea extract) was used as the control. As a result, the residual activity of the enzyme decreased dose-dependently, which shows that this herbal tea may prevent the α-glucosidase from working at certain concentrations. The range of effects is evident: for example, the extraction dosages (from May) varied from 0.002 mg/mL to 100 mg/mL, resulting in residual percentages from 76.15% to 1.97%. whereas the 0.2 and 100 mg/mL tea samples in March protected 97.96% and removed all of the α-glucosidase, respectively. For this reason, we compared the suppression capacity with the IC_50_ value, which is listed in Table 5. Tea from March inhibited 50% of the enzymes at a concentration of 5.89 ± 0.29 mg/mL, and that from January occurred at a concentration of 20.65 ± 0.19 mg/mL. The acarbose group was used as the positive control, of which the α-glucosidase inhibitory effect (50%) was 4.27 ± 0.17 mg/mL. Ideally, when other IC_50_ values displayed no significant difference within the year, the inhibitory activity of March and December showed peak values, and March is roughly equal to typical acarbose, and the outcome indicates that the best among the 12 tea samples possessed a good inhibition effect on α-glucosidase.

#### 2.4.2. Lipase Repression

Lipase, primarily produced in the pancreas, hydrolyses lipids to form fatty acids so that they can be absorbed in the human digestive system. Digestion of fat is a prerequisite for its uptake and the inhibition of pancreatic lipase, the most important enzyme responsible for the digestion of dietary triglycerides, can be used to manage weight and potentially reduce obesity [30,31]. With respect to the inhibitory indicator, the commercial drug orlistat is generally selected as reference [32]. Combined with the traditional pharmacological functions, drinking Hawk tea can be useful to treat obesity as a dietary supplement. In this study, we measured the repression effect of our samples on porcine pancreas lipase, contrasted the results with those of the use of orlistat, and compared the results. As shown in Table 5, orlistat produced the lowest IC_50_ (0.006 mg/mL), whereas March and November produced the tea with the strongest suppression on lipase, with IC_50_ values of 17.02 ± 2.65 and 12.49 ± 0.81 mg/mL, respectively. The other months produced an average distribution, except for September, which showed a 50% inhibition of lipase with a highest concentration of 42.60 ± 1.32 mg/mL, indicating the weakest inhibition.

Lipase, primarily produced in the pancreas, hydrolyses lipids to form fatty acids so that they can be absorbed in the human digestive system. Digestion of fat is a prerequisite for its uptake and the inhibition of pancreatic lipase, the most important enzyme responsible for the digestion of dietary triglycerides, can be used to manage weight and potentially reduce obesity [30,31]. With respect to the inhibitory indicator, the commercial drug orlistat is generally selected as reference [32]. Combined with the traditional pharmacological functions, drinking Hawk tea can be useful to treat obesity as a dietary supplement. In this study, we measured the repression effect of our samples on porcine pancreas lipase, contrasted the results with those of the use of orlistat, and compared the results. As shown in Table 5, orlistat produced the lowest IC_50_ (0.006 mg/mL), whereas March and November produced the tea with the strongest suppression on lipase, with IC_50_ values of 17.02 ± 2.65 and 12.49 ± 0.81 mg/mL, respectively. The other months produced an average distribution, except for September, which showed a 50% inhibition of lipase with a highest concentration of 42.60 ± 1.32 mg/mL, indicating the weakest inhibition.

### 2.5. Correlation Analysis

As shown in Table 6, none of the antioxidant assays showed a close correlation with the TPC, which varied from 10.91 ± 0.5 to 21.92 ± 1.87 mg/g (GAE/DW). The TFC significantly affected the oxidation resistance according to the determination coefficients (*p* < 0.05) 0.916, 0.935, and 0.955 of DPPH, ABTS and reducing power, respectively. Whereas the total phenolics in our exploration seemed to be insignificant (R^2^ = 0.504 for DPPH, 0.238 for ABTS, and 0.417 for ferric reducing potential).

### 2.6. Hierarchical Cluster Analysis

To examine the optimum combination of the chemicals amounts to improve the tested pharmacological effects, and finally to determine the best picking time, the squared Euclidean distance [33] was adopted for hierarchical clustering. The outcome is shown in Figure 3. 

Simply, the 12 samples are clustered into three groups (I, II, and III), where group I is composed of four samples, II contains two, and group III includes six different months. May, December, June, August belonged to group I for their similar weak activities and effective constituents, October and November formed group II, and the other times of the year were classified as III, which was moderate in terms of active ingredients, antioxidants and inhibitors of enzyme. Within the squared Euclidean distance of 5, samples were then sorted into four classes, where January and February were distinct from March, July, September and April. In total, the six months of the last group showed no difference. In terms of the quantitation and efficacy trial, the collection time in group II, which contained beverage materials from October and November, may be preferable in terms of the harvest months.

## 3. Discussion

For active substances in ripe leaves of *Litsea coreana* Levl. var. lanuginose, both total components of flavonoid and phenolic compounds were detected for the fundamental analysis. The TFC and TPC change from month to month, and relatively good quantities were found in September to November. Flavonoids are the most important quality-related compounds and the main group of polyphenols and they are important antioxidants due to their high redox potential [34,35]. Monomeric flavonoids in this study were separated and quantitatively analyzed based on preliminary data and comparison with compound references. The seven determined characteristic constituents were found in all samples, indicating that the harvest time did not influence the similarity in composition. This finding is partly in agreement with Ma et al. [24], but with differences mainly in the quantities, which vary with irregular trends for different months.

The production and scavenging of free radicals maintains a dynamic balance in the human body [6]. Hawk tea is broadly accepted because of its high antioxidant activity [36]. In line with our previous work, Hawk tea is effective in the DPPH, ABTS, and reducing power assays, and the three activities in the different tests change following similar trends. The numerical differences might reflect differences in the ability of antioxidant compounds to act against the different radicals present or formed during each specific reaction. Taken together, the results illustrate that Hawk tea is an exogenous plant antioxidant, and that April, September, October and November produced better antioxidants.

Based on its historical Chinese use, the extensive acceptability of Hawk tea can be partly attributed to the benefits of promoting digestion and resolving greasy food, and preventing food spoilage, abdominal distension and sunstroke [37]. One therapeutic approach to decrease postprandial hyperglycemia is to suppress the production and/or absorption of glucose from the gastrointestinal tract through the inhibition of α-glucosidase or α-amylase [38]. Much research has focused on glycosidase inhibitors to control hyperglycemia. To deal with these health ailments, synthetic drugs such as acarbose are often used but have several adverse side effects, such as abdominal discomfort, diarrhea, flatulence, and hepatotoxicity [31]. For Hawk tea, α-glucosidase inhibition and lipase repression tests were conducted to compare the activity of reducing blood sugar and lipids, and the presented results are promising. Crude extract of this tea is a complex form of bioactive constituents, other chemicals such as organic acids, essential oils, saponins, and polysaccharides were found in *Litsea coreana* [5,9,21,39], which can potentially act on the enzymes mentioned in this research.

In line with previous findings, Hawk tea could likely be used as a natural material to develop α-glucosidase inhibitors. Xiao et al. [15] comparatively analyzed the antioxidant, antibacterial, in vivo, and in vitro hypoglycemic effects of different product regions and active fractions of Hawk tea. Qin et al. [11] explored the bioactivities of the essential oil of this tea (from six habitats). In contrast, firstly, we considered the monthly variation, and our integrated results show that, this drink produced in March and August comparatively presented the greatest inhibiting effect. Secondly, the lipase suppression of Hawk tea has rarely been investigated, but was reported in this work. Hawk tea is a more advisable edible and therapeutic crude drug for managing obesity as a daily and reasonably- priced drink, despite the fact that orlistat can be more efficient. Obesity was found to be partly associated with low antioxidant status [31], thus, the connection of antioxidants with obesity control help manage chronic diseases, prompting the use of Hawk tea.

The correlation coefficients present the importance of TFC in DPPH and ABTS scavenging assays, along with ferric reducing capability. This finding is in accordance with the early reports [23,24] analyzing the major ingredients and bioactivities of Hawk tea’s phenolic compounds. The HCA result was obtained by the comprehensive data of TPC, TFC, total monomeric flavonoid contents, antioxidant, and enzyme inhibition assays. Twelve harvest times were divided into three groups: the best, moderate and poor months. October and November are the best choices in consideration of the active compounds and bioactivities.

The measured results varied among the 12 months. For most plants, the accumulation of secondary metabolites is affected by many factors. Li et al. [40] concluded that phytochemicals can be affected by the growing period, which explains the diversity of the component contents in different seasons. As a kind of metabolites, phytochemicals can also be influenced by external changes such as monthly light irradiation, moisture, and temperature, etc. For instance, Regvar et al. [41] found a specific increase of quercetin concentration in *Fagopyrum. esculentum* when exposed to enhanced ultraviolet (UV) irradiation. The variation in secondary metabolites accumulation eventually discriminated the pharmacological effects among different months.

Considering its favorable biological effects, mature leaf Hawk tea deserves full exploration, and the present study provides a basis for its development and utilization. Flavor and appearance are also important in Chinese tea culture, and March and April used to be the default picking times. However, the evidence is that tea from September to November should not be ignored as the plants picked in these months have the best antioxidant and health qualities; July and September are alternative collection periods. As a whole, Hawk tea possesses a combination of the merits above could become a more valuable resource. In this study, we only focused on the dominant active chemicals and bioactivities, and more sensory indicators will be introduced in our following works.

## 4. Materials and Methods

### 4.1. Plant Material and Regents

The samples from the 12 months of 2017, were picked from PengShui County, Chongqing, China and authenticated by D.Z. (Chongqing Medical University, Chongqing, China). Rutin, quercetin, kaempferol, astragalin, quercetin-3-d-galactoside, quercetin-3-β-d-glucoside, and quercetin-3-rhamnosidee (purity 99.0% each) were purchased from the Natl. Inst. (Beijing, China). 1,1-Diphenyl-2-picrylhydrazyl (DPPH), butylated hydroxytoluene (BHT), and 2,2-azinobis (3-ehtylbenzothiazolin-6-sulfinic acid) diammonium salt (ABTS) were purchased from Aladdin Co. (Shanghai, China). Trichloroacetic acid, ferric chloride and potassium ferricyanide were purchased from Sinopharm Chemical Reagent Co., Ltd. (Shanghai, China). Yeast α-glucosidase (EC 3.2.1.20) and porcine pancreatic lipase (EC 3.1.1.1) were purchased from Sigma-Aldrich (St. Louis, MO, USA). *p*-Nitrophenyl-α-d-glucopyranoside (p-NPG) and *p*-nitrophenyl palmitate were obtained from Sigma-Aldrich Chemicals Pvt. Ltd. (Bangalaru, India). Chromatographic grade methanol and acetonitrile were purchased from Alltech Scientific (Beijing, China). The remaining reagents were of analytical grade and were purchased from Chongqing Chemical Works Co., Ltd. (Chongqing, China).

### 4.2. Sample Preparation

The samples were air dried in shade and cut into pieces. Ten grams of the sample powder were immersed using 50 mL of 70% (*v/v*) aqueous ethanol solution, mixed for 2 h under room temperature, followed by treatment twice with ultrasound (300 w, 50 Hz) for 60 min. The pooled extract was filtered through a 0.22 μm filter and the solvent was removed using a rotary evaporator under 50 °C; a suitable volume of solvent was added to maintain a concentration of 0.1 g/mL, and then stored in a dark place for further analysis. All samples were tested in triplicate.

### 4.3. Determination of TFC

The colorimetric method described by Awah et al. [42] with slight changes was adopted to measure the TFC of the tea for the 12 months. We added the extract (2 mL) to a 25 mL brown volumetric flask with 8 mL of 50% ethanol, and then 1 mL of 5% NaNO_2_ solution was added and evenly mixed. Six minutes later, 1 mL of 10% Al(NO_3_)_3_ solution was added and the mixture was allowed to stand for 6 min, then 10 mL of 4% NaOH solution was pipetted into the mixture. The resulting mixture was diluted with 3 mL of double distilled water and left for 15min before detection. The absorbance at 510 nm was detected using a spectrophotometer (UV-1780, Shimadzu, Kyoto, Japan). For the present study, the standard curve was determined with rutin, and the TFC was expressed as mg rutin equivalent (RE) per 1.0 g dry sample weight. All experiments were finished in triplicate.

### 4.4. Determination of TPC

The determination of TPC was based on the modified colorimetric method in Hajimahmoodi et al. [43]. The sample (0.5 mL) was mixed with 4.5 mL of ethanol in a 25 mL brown volumetric flask. Afterwards, 2 mL of a 0.3% sodium dodecyl sulphate solution, 2 mL of the pooled solution of 0.5% potassium ferricyanide and 1% ferric chloride (1:1) were added. After being fully mixed, the solutions were allowed to stand for 5 min in the dark. Then, 0.1 mol/L hydrochloric acid (HCL) solution was added produce a final volume of 25 mL, followed by thoroughly shaking and standing for another 20 min in the dark. The absorbance at 760 nm was tested using a UV–visible spectrophotometer (provided in Section 4.3). The total phenolic content was estimated using a standard curve prepared with gallic acid and is expressed as milligrams of gallic acid equivalents (GAE) per gram of dry tea sample. All samples were tested in triplicate.

### 4.5. Quantification of Flavonoid Monomers and Method Validation

Flavonoids were tested by HPLC (LC-20AD, Shimadzu) with a photodiode array detector (DAD; Shimadzu). A Hypersil ODS2 column (5 μm, 4.6 mm × 250 mm i.d., SinoChrom, Dalian, China) with the mobile phase, aqueous 0.1% formic acid solution (A) and acetonitrile (B) and a flow rate of 1 mL/min with a 10 μL injection volume were used, and the column oven temperature was set to 30 °C. For better isolation, the gradient program was set as follows: 0–38 min, 11–20% B; 38–66 min, 20–40% B; 66–66.01 min, 40–11% B; and 66.01–70 min, 11% B. The detection wavelength was set to 350 nm.

#### 4.5.1. Linearity, Limits of Detection and Limits of Quantification

Ten milliliters of standard solution mixture in series concentrations were injected into the HPLC system, and the chromatography was recorded. The calibration curves were established using six different concentrations of chemical markers, plotted by the peak area (Y) of analytes against concentration (X). The limits of detection (LOD) and limits of quantification (LOQ) were measured based on the signal-to-noise ratio (SNR) of 3 and 10, respectively. Each concentration was analyzed in triplicate.

#### 4.5.2. Precision, Accuracy, Repeatability and the Stability Tests

To verify the precision, 10 μL of the six -reference mixture were continuously detected for peak areas and retention times. For the accuracy (recovery), repeatability and the stability tests (0, 2, 4, 8, 12, and 24 h), 10 μL of the sample solution from November were used. All the analytes were analyzed for six times, the results were reflected by the RRT and RPA, and the RSD was used to evaluate the methods. The recovery test calculation was according to the following formula:recovery (%) = (found amount − original amount)/spiked amount × 100%(1)

### 4.6. Antioxidant Activity Evaluation

#### 4.6.1. Scavenging Effect on DPPH Radical

The assay was carried out by a known method [44] with some alterations. Concisely, the sample solution (200 μL) in an Eppendorf tube was blended with a methanol solution of DPPH (800 μL). Ultrapure water was used as a control instead of the sample solution. The mixture was shaken effectively and incubated in the dark for exactly 1 h, then the absorbance was read at 517nm by the UV-vis plate reader (provided in Section 4.3). The radical scavenging activity was calculated using the equation:%scavenging = [(A_control_ − ∆A_sample_)/A_control_] × 100(2)
where A_control_ is the absorbance of the the control (sample replaced by PBS), and ∆A_sample_ is the test sample. BHT was used as the positive control, and IC_50_ values represent the extracts and controls scavenging 50% of DPPH radicals. All the IC_50_ values were calculated, where the values represented the scavenged 50% of DPPH radicals of aqueous extracts and positive controls.

#### 4.6.2. Scavenging Activity against ABTS radicals

The scavenging activity against ABTS was tested according to our previous method [12]. Briefly, a certain amount of MnO_2_ was reacted with 7 mM ABTS to produce ABTS radical cations, then stored and incubated in the dark at room temperature for 16 h. PBS (pH 7.4) was added to dilute the mixture until the absorbance reached 0.70 (±0.02) at 734 nm. Fifty microliters of sample or control in various concentrations were added to 3.0 mL of the ABTS reaction system, fully shaken, placed for 6 min at room temperature in the dark, and then the absorbance was determined at 734 nm. The ABTS radical scavenging activity was calculated using the formula:%scavenging = [(A_control_ − ∆A_sample_)/A_control_] × 100(3)
where A_control_ is the absorbance of the control (sample replaced by PBS), and ∆A_sample_ represents the test sample. The results are shown as IC_50_ to facilitate comparison.

#### 4.6.3. Reducing Power

The measurement in Wang et al. [45] with a slight modification, was conducted to detect the reducing power. The sample to be determined (2 mL) in a serial concentration was mixed with K_3_Fe(CN)_6_ (1%, *w/v*, 2.5 mL). The mixture was incubated at 50 °C for 20 min, then 2.5 mL of 10% trichloroacetic acid was added, followed by centrifugation at 3000 rpm. for 10 min. After, a 2.5 mL aliquot of the upper layer was mixed with ultrapure water (2.5 mL) and 0.5 mL of 0.1% FeCl_3_. The absorbance at 700 nm against a blank was recorded 10 min later. BHT was used as a reference.

### 4.7. Enzyme Inhibition

#### 4.7.1. α-Glucosidase Inhibition Assay

The evaluation of α-glucosidase inhibition was performed as in Apostolidis et al. [46] with a minor modification. A mixture of 50 μL Hawk tea or acarbose solution and 100 μL of 0.1M phosphate buffer (pH 6.9) containing α-glucosidase solution (1.0 U/mL) was incubated in 96-well plates at 25 °C for 10 min. After preincubation, 50 μL of 5 mM p-NPG solution in 0.1 M phosphate buffer (pH 6.9) were added to each well at timed intervals. The reaction mixtures were incubated at 25 °C for 5 min. Before and after incubation, absorbance was recorded at 405nm by a micro-plate reader (Synergy HTX, BioTek, Winoski, VT, USA) and compared to that of the control, which was 50 μL buffer solution in place of the extract. The α-glucosidase inhibitory activity is expressed as an inhibition percentage and calculated as follows:%inhibition = [(∆A_control_ − ∆A_sample_)/(∆A_control_)] × 100(4)
where ∆A_control_ and ∆A_sample_ are the absorbance of the sample and the control, respectively. The inhibitory activity is expressed as the half-maximal inhibitory concentration (IC_50_).

#### 4.7.2. Lipase Inhibition

Lipase activity was assayed base on Costamagna et al. [14] with slight changes. The ability to inhibiting lipase was assessed by measuring the enzymatic hydrolysis of *p*-nitrophenyl palmitate to *p*-nitrophenol in a microplate reader (provided in Section 4.7.1) at 400 nm. Lipase solution (1.2 mg/mL) was mixed with the tea extract (final concentration between 1.00 and 66.67 mg/mL) and pre-incubated on ice for 5 min. The reaction mixture contained 330 μL of sodium phosphate buffer 0.1 M (pH 7) with 20 μL 10 mM *p*-nitrophenyl palmitate. The enzyme reaction was started by adding 50 μL of the lipase/tea extract solution into the reaction mixture, and incubated at 37 °C for 20 min. The absorbance at 400 nm was recorded, and the inhibitory activity was calculated as follows:%inhibition = [(∆A_control_ − ∆A_sample_)/(∆A_control_)] × 100(5)
where ∆A_control_ is the absorbance of the control (blank, without extract) and ∆A_sample_ is the absorbance in presence of the extract. Orlistat was selected as the positive control, and IC_50_ values represent the lipase inhibition of 50% of inhibitors.

### 4.8. Statistical Analysis

Statistical analysis was performed by SPSS 20.0 (SPSS Inc., Chicago, IL, USA). Data are expressed as means ± standard deviation (SD). A bivariate correlate analysis was used to determine the correlation. One-way analysis of variance (ANOVA) was used to evaluate differences, and *p* < 0.05 was regarded as statistically significant.

## 5. Conclusions

In this study, we determined the total flavonoids and phenolic acids from the extract of Hawk tea leaves. For individual flavonoids, including hyperin, isoquercitrin, trifolin, quercitrin, astragalin, quercetin, and kaempferol, there was no obvious difference in the 12 months in terms of the composition, but the contents had different proportions. The DPPH, ABTS, and ferric reducing power reflect the antioxidant ability. The α-glucosidase and lipase inhibition capabilities, which play important roles in human health, were measured. Variations in these abilities were obvious according to the results. By comprehensively analyzing the chemical contents and bioactivity results, HCA groups were obtained to determine the best months for tea harvesting. We conclude that October and December are the best months. More bioactivity and indicators will be determined in our further steps, hoping to optimize the picking conditions of this beneficial herb, provide a more comprehensive quality evaluation method, and improve the economic profits of rural areas.

## Figures and Tables

**Figure 1 molecules-24-00657-f001:**
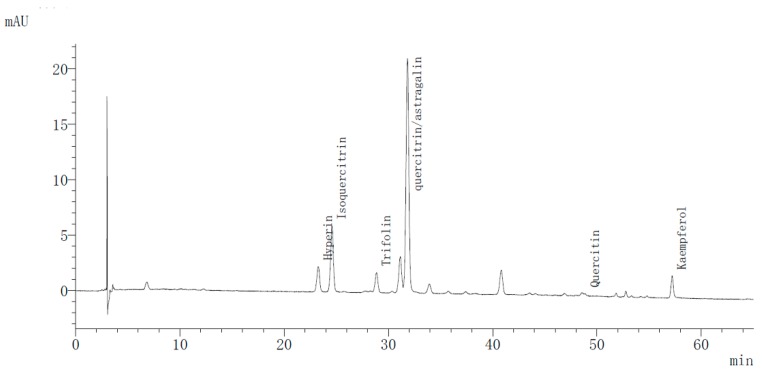
High performance liquid chromatography profile of tea extract from Nov at 350 nm.

**Figure 2 molecules-24-00657-f002:**
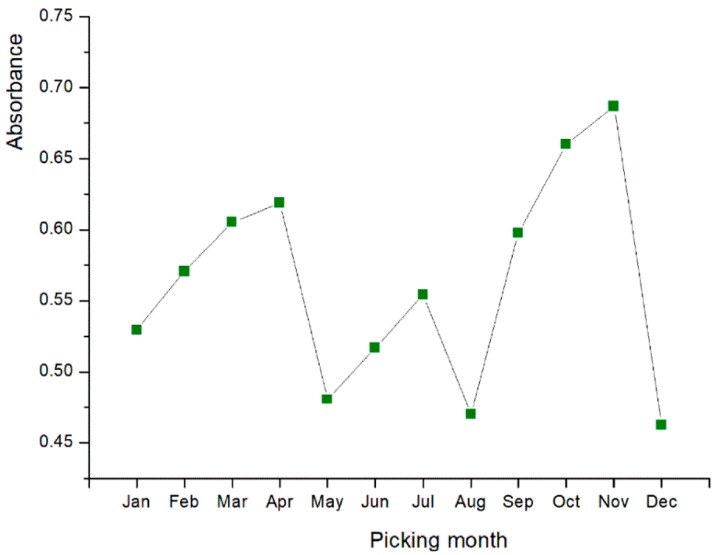
The absorbance variation of 12 sample solutions (5 mg/mL) in the radical reducing power reaction.

**Figure 3 molecules-24-00657-f003:**
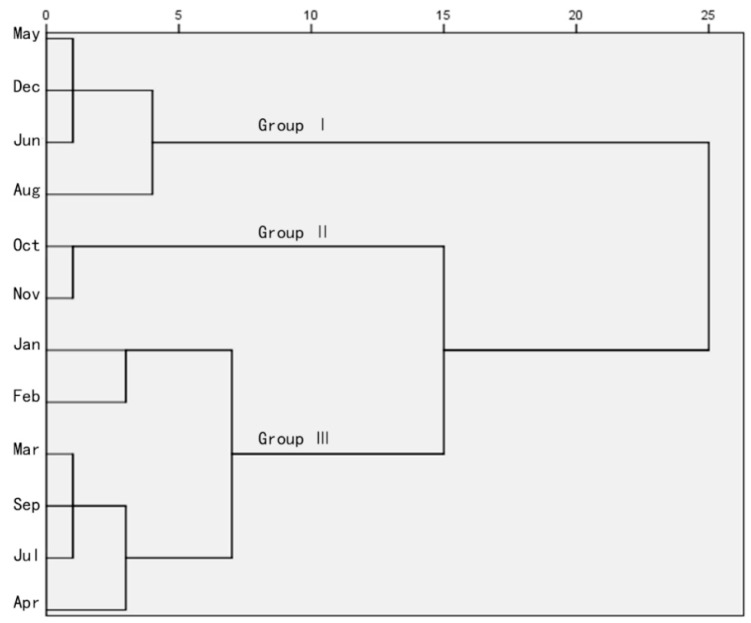
Cluster analysis diagram based on ingredient contents and bioactivities.

**Table 1 molecules-24-00657-t001:** Total flavonoids and total phenols contents of samples from twelve months.

Month	Total Flavonoids (RE mg/g DW)	Total Phenolic Acids (GAE mg/g DW)
January	8.00 ± 0.48 ^cd^	12.37 ± 0.44 ^e^
February	8.40 ± 0.91 ^cd^	14.50 ± 1.13 ^d^
March	9.53 ± 0.32 ^bc^	14.99 ± 0.67 ^d^
April	9.31 ± 0.51 ^bc^	17.17 ± 0.72 ^c^
May	7.40 ± 0.88 ^d^	12.97 ± 0.22 ^e^
June	6.79 ± 0.29 ^d^	12.51 ± 0.41 ^e^
July	8.86 ± 0.55 ^c^	16.51 ± 0.15 ^c^
August	6.91 ± 0.38 ^d^	21.92 ± 1.87 ^a^
September	10.14 ± 0.60 ^b^	18.97 ± 0.12 ^b^
October	11.75 ± 0.65 ^a^	16.53 ± 0.28 ^c^
November	11.57 ± 0.83 ^a^	19.89 ± 0.78 ^b^
December	6.50 ± 0.48 ^d^	10.91 ± 0.5 ^f^

Values are expressed as mean ± standard deviation; different superscript lowercase letters denote statistically significant difference (*p* < 0.05).

**Table 2 molecules-24-00657-t002:** Contents of hyperin, isoquercitrin, quercitrin and astragalin, quercetin and kaempferol of different samples (in μg/g).

Month	Hyperin	Isoquercitrin	Trifolin	Quercitrin/Astragalin	Quercetin	Kaempferol	Total Contents
Jan	35.98	90.67	17.57	232.63	6.96	7.48	391.29
Feb	28.59	102.62	17.58	401.43	5.83	7.99	564.04
Mar	41.97	104.1	20.49	269.80	5.73	7.80	449.89
Apr	32.64	90.86	14.82	311.28	5.62	8.26	463.48
May	31.82	80.28	17.91	301.28	5.55	8.86	445.7
Jun	28.63	93.85	18.77	369.04	5.62	6.21	522.12
Jul	43.66	102.95	22.91	346.20	6.56	4.33	526.61
Aug	29.48	87.27	15.59	354.27	5.38	15.62	507.61
Sep	42.01	137.35	17.24	461.90	5.57	9.19	673.26
Oct	36.36	104.7	21.65	379.85	5.82	12.73	561.11
Nov	49.11	122.99	45.43	447.32	11.01	26.57	702.43
Dec	32.97	86.02	20.33	277.99	9.39	17.34	444.04

**Table 3 molecules-24-00657-t003:** Calibration curve, LOD and LOQ, and linear range for six standards.

Analytes	Regression Equation	R^2^	LODs (μg/mL)	LOQs (μg/mL)	Linear Range (μg/mL)
Hyperin	y = 19587x − 1811.5	1	0.09	0.278	0.32–206.3
Isoquercitrin	y = 19291x − 1922	1	0.09	0.27	0.43–273.8
Trifolin	y = 16353x − 2087.6	1	0.09	0.26	0.57–362.5
Quercitrin/Astragalin	y = 19270x − 2705.2	1	0.11	0.33	0.47–302.5
Quercetin	y = 24303x − 6535.9	0.9999	0.18	0.53	0.39–252.5
Kaempferol	y = 29685x − 2772.3	0.9999	0.06	0.18	0.20–126.3

**Table 4 molecules-24-00657-t004:** DPPH, ABTS assays results (in mg/mL) of Hawk tea from 12 months.

Month	DPPH IC_50_	ABTS IC_50_
January	0.21 ± 0.001 ^g^	3.14 ± 0.03 ^e^
February	0.19 ± 0.001 ^e^	2.80 ± 0.07 ^d^
March	0.16 ± 0.005 ^d^	2.77 ± 0.01 ^d^
April	0.15 ± 0.003 ^c^	2.30 ± 0.01 ^d^
May	0.20 ± 0.003 ^f^	4.69 ± 0.03 ^g^
June	0.20 ± 0.006 ^f^	4.00 ± 0.18 ^f^
July	0.16 ± 0.006 ^d^	2.70 ± 0.09 ^d^
August	0.20 ± 0.003 ^f^	7.51 ± 0.29 ^h^
September	0.16 ± 0.008 ^d^	2.29 ± 0.07 ^bc^
October	0.15 ± 0.007 ^bc^	1.92 ± 0.02 ^bc^
November	0.14 ± 0.001 ^b^	2.05 ± 0.03 ^bc^
December	0.20 ± 0.008 ^f^	4.52 ± 0.18 ^g^
BHT	0.04 ± 0.001 ^a^	0.60 ± 0.01 ^a^

Values are expressed as mean ± standard deviation; different superscript lowercase letters denote statistically significant differences (*p* < 0.05).

**Table 5 molecules-24-00657-t005:** Results of inhibition on α-glucosidase and lipase.

Month	α-Glucosidase IC_50_ (mg/mL)	Lipase IC_50_ (mg/mL)
January	20.65 ± 0.19 ^k^	35.40 ± 0.23 ^cd^
February	14.72 ± 0.02 ^i^	33.06 ± 1.45 ^cd^
March	5.89 ± 0.29 ^b^	17.02 ± 2.65 ^b^
April	9.58 ± 0.03 ^e^	29.21 ± 1.41 ^cd^
May	8.19 ± 0.09 ^d^	32.72 ± 1.66 ^cd^
June	9.35 ± 0.06 ^e^	31.18 ± 1.52 ^d^
July	10.61 ± 0. 03 ^fg^	34.20 ± 0.42 ^d^
August	7.36 ± 0.37 ^c^	36.11 ± 2.41 ^d^
September	13.14 ± 0.05 ^h^	42.60 ± 1.32 ^e^
October	10.19 ± 0.29 ^f^	34.80 ± 1.46 ^d^
November	10.89 ± 0.05 ^g^	12.49 ± 0.81 ^b^
December	18.13 ± 0.02 ^j^	36.86 ± 1.24 ^de^
Acarbose	4.27 ± 0.17 ^a^	ND
Orlistat	ND	0.0006 ^a^

Values are expressed as mean ± standard deviation; different superscript lowercase letters denote statistically significant differences (*p* < 0.05).

**Table 6 molecules-24-00657-t006:** The correlation between DPPH, ABTS, ferric reducing capacity with total flavonoids and total phenols.

Component	DPPH	ABTS	Reducing Power
Total flavonoids	R^2^ = 0.916 **	R^2^ = 0.935 **	R^2^ = 0.955 **
Total phenols	R^2^ = 0.504	R^2^ = 0.238	R^2^ = 0.417

** Extremely significant correlation, *p* < 0.01.

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
