# Peer review of "Active Components, Antioxidant, Inhibition on Metabolic Syndrome Related Enzymes, and Monthly Variations in Mature Leaf Hawk Tea"

_molecules, 2019, doi:10.3390/molecules24040657_

Round 1
Reviewer 1 Report
Leaves from Litsea coreana have been used as beverage in Southwestern China. The authors investigated the chemical composition and anti-oxidant activity of Litsea coreana leaves harvested in different months during 2017. The manuscript has informational value. I recommend its publication after major revision.
1. The advantages and disadvantages of the present manuscript should be described with comparison to those by Xiao et al (2017) (DOI: 10.3390/molecules22101622) and Qin et al (2018) (DOI: 10.1016/j.indcrop.2018.07.035).
2. Data in Figure 1 should be expressed as a Table.
3. The title should be re-written.
4. The English writing is quite poor and should be polished by a native English speaker. For example:
Lines 83-84: However, the results of December that 6.50 ±0.48 and 10.91 ±0.5 mg RE/ g DW respectively for TFC and TPC manifests its inferior in content.
Lines 110-113: [20] reported the protective effect of quercitrin against APAP-induced hepatotoxicity through enhancement of activity and expression of antioxidant enzymes, and [21] claimed that astragalin may be a potent agent antagonizing endotoxin-induced oxidative stress leading to airway dysfunction and inflammation.
Line 208: Figure 4 should be replaced by Table 4.
Lines 208-210: …all of the oxidant assays did not show a close relation with the 208 TPC, which varies from 10.91 ±0.5 to 21.92 ±1.87 mg GAE / g DW, and sample from May makes up the highest place.
Lines 231-232: Combined with … function.
Line 239: …phenolic acid in the materials we discussing is out of strong role
Line 244: …in a hole year.
Line 246-247: Extensive acceptability of hawk tea stems from classical China can partly attribute to the benefit of promoting digestion and resolving greasy food,…
Line 269: …aside from samples from the sample from Chongqing…
In addition, this manuscript is lacking in explanation about the reasons leading to the differences in the chemical composition and in the bioactivities.
Author Response
Point 1: The advantages and disadvantages of the present manuscript should be described with comparison to those by Xiao et al (2017) (DOI: 10.3390/molecules22101622) and Qin et al (2018) (DOI: 10.1016/j.indcrop.2018.07.035).
Response 1: The advantages and disadvantages of our revised manuscript have been described in line 280-290, and also shown as follows:
As with previous findings, Hawk tea is likely to be used as natural materials to develop α-glucosidase inhibitors. Xiao et al. comparatively analyzed the antioxidant, antibacterial, in vivo, and in vitro hypoglycemic effects of different product regions and active fractions of Hawk tea. Qin et al. explored the bioactivities of the essential oil of this tea (from six habitats). In contrast, firstly, we considered the monthly variation, and our integrated results show that, this drink produced in March and August comparatively presented the greatest inhibiting effect. Secondly, the lipase suppression of Hawk tea has rarely been investigated, but was reported in this work. Hawk tea is a more advisable edible and therapeutic crude drug for managing obesity as a daily and reasonably- priced drink, despite the fact that orlistat can be more efficient. Obesity was found to be partly associated with low antioxidant status [31], thus, the connection of antioxidants with obesity control help manage chronic diseases, prompting the use of Hawk tea.
Point 2: Data in Figure 1 should be expressed as a Table.
Response 2: Figure 1 was replaced by Table 2, in page 4.
Point 3: The title should be re-written.
Response 3: The title “The optimization of hawk tea picking time based on the effects on oxidant resistance and enzymes involved in metabolic syndrome” is replaced by “Active components, antioxidant, inhibition on metabolic syndrome related enzymes, and monthly variations in mature leaf Hawk tea.”
Point 4: The English writing is quite poor and should be polished by a native English speaker. For example:
·Lines 83-84: However, the results of December that 6.50 ±0.48 and 10.91 ±0.5 mg RE/ g DW respectively for TFC and TPC manifests its inferior in content;
Response: The sentence has been replaced by “The TFC and TPC for December were 6.50 ± 0.48 mg RE/g DW and 10.91 ± 0.5 mg RE/g DW, respectively, demonstrating inferiority in terms of that month’s polyphenol content.” (lines 99-100)
·[20] reported the protective effect of quercitrin against APAP-induced hepatotoxicity through enhancement of activity and expression of antioxidant enzymes, and [21] claimed that astragalin may be a potent agent antagonizing endotoxin-induced oxidative stress leading to airway dysfunction and inflammation.
Response: The sentence has been replaced by “Truong et al. [25] reported that quercitrin can act against acetaminophen (APAP)-induced hepatotoxicity through the enhancement of the activity and expression of antioxidant enzymes, and Cho et al. [26] reported that astragalin may be a potent agent in antagonizing endotoxin-induced oxidative stress, which can lead to airway dysfunction and inflammation.” (lines 121-125)
·Line 208: Figure 4 should be replaced by Table 4.
Response: Figure 4 has been replaced by Table 6 (line 224)
·Lines 208-210: …all of the oxidant assays did not show a close relation with the 208 TPC, which varies from 10.91 ±0.5 to 21.92 ±1.87 mg GAE / g DW, and sample from May makes up the highest place.
Response: The sentence has been replaced by “none of the antioxidant assays showed a close correlation with the TPC, which varied from 10.91 ± 0.5 to 21.92 ± 1.87 mg/g (GAE/DW).” (lines 224-225)
·Lines 231-232: Combined with … function.
Response: The sentence has been replaced by “In terms of the quantitation and efficacy trial, the collection time in group II, which contained beverage materials from October and November, may be preferable in terms of the harvest months.” (lines 244-246)
·Line 239: …phenolic acid in the materials we discussing is out of strong role
Response: This sentence has been removed.
·Line 244: …in a hole year.
Response: This sentence has been removed.
·Line 246-247: Extensive acceptability of hawk tea stems from classical China can partly attribute to the benefit of promoting digestion and resolving greasy food…
Response: This sentence has been replaced by “Stemming from historical China, the extensive acceptability of Hawk tea can be partly attributed to the benefit of promoting digestion and resolving greasy food, and preventing food spoilage, abdominal distension and sunstroke.” (lines 268-270)
·Line 269: …aside from samples from the sample from Chongqing…
Response: This sentence has been removed.
Point 5: This manuscript is lacking in explanation about the reasons leading to the differences in the chemical composition and in the bioactivities.
Response 5: This part is discussed in lines 298-305, and shown as follows:
The measured results varied among the 12 months. For most plants, the accumulation of secondary metabolites is affected by many factors. Li et al. [41] concluded that phytochemicals can be affected by the growing period, which explain the diversity of the component contents in different seasons. As a kind of metabolites, phytochemicals can also be influenced by external changes such as monthly light irradiation, moisture, and temperature, etc. For instance, Regvar et al. [42] found a specific increase of quercetin concentration in Fagopyrum. esculentum when exposed to enhanced ultraviolet (UV) irradiation. The variation in secondary metabolites accumulation eventually discriminated the pharmacological effects among different months.

Reviewer 2 Report
In this manuscript, author studied the anti-oxidant activity and inhibition effect of several enzyme of the hawk tea in various season. There are some unclear points. Please consider below.
1. P4 Figure 1
Flavonoid contents of February tended to higher. However, total flavonoids contents was low level compared with September and November. February seems to exception case. It should be discussed the reason.
2. P6: anti-oxidation activity
The results of three anti-oxidant activity experiments were different. How it should be consider these difference?
3. Table 3
Inhibitory effects of alpha-glucosidase and lipase activity is normally related to polyphenol content. However, in this experiments, the result did not depend to polyphenol content. For example, May was strong inhibitory effect of alpha-glucosidase activity. However, low anti oxidative activity and low content of total phenolic compound. What do you think this point?
4. As a whole, I think this manuscript is topic is unclear. For example, Figure 2 and Figure 5 are far from topic of this manuscript title. Result section contained discussion, and discussion section contained result. It should be re consisted the sentence.
Minor points
1. Page 6 line 139
November→December?
2. Page8 line183 to 190, 202 to 206, 212 to 216
This sentence is discussion. It is not results. It should be move to discussion section.
3. P9 line237
Litsea coreane H →Scientific name should be written by italic.
3. P12 line 321
It should be described HPLC system name, flow speed, injection volume etc.
Author Response
Point 1: Figure 1 Flavonoid contents of February tended to higher. However, total flavonoids contents was low level compared with September and November. February seems to exception case. It should be discussed the reason.
Response 1: This question does worth thinking about. This study we focused on total flavonoid contents (TFC), phenolic contents and contents of characteristic flavonoid monomers. Our redetermination results are anastomotic with before. Then we analysed the correlation between TFC and the total monomers contents, the determination coefficient R2=0.0621*, while we remove the values of February and analyse the other eleven samples, the R2 is 0.0638*. The tiny difference proves that February is actually normal even it seems obstructive. It also can be predicted that there exist other flavonoids in this tea. Additionally, detection by colorimetric method may influenced by extra interferences.
Point 2: The results of three anti-oxidant activity experiments were different. How it should be consider these difference?
Response 2: In this round of modification, we have applied another several days to find out the problem leading to the difference in our preliminary manuscript. In our retest, we increased the sample size and renew all of the regents. Via analysing correlations among the three assays and with active components, and eventually obtain more reasonable results. The determination coefficients are shown in the table below.
| Assays | DPPH | ABTS | Reducing power | |
| DPPH | R2=1 | R2= 0.877** | R2=0.917** | |
| ABTS | R2= 0.877** | R2=1 | R2= 0.942** | |
| Reducing power | R2=0.917** | R2=0.942** | R2=1 | |
Point 3: Inhibitory effects of alpha-glucosidase and lipase activity is normally related to polyphenol content. However, in this experiment, the result did not depend to polyphenol content. For example, May was strong inhibitory effect of alpha-glucosidase activity. However, low anti oxidative activity and low content of total phenolic compound. What do you think this point?
Response 3: Traditional Chinese herb works through multi-target and multi-pathway. In this study, the crude extract was used to investigate the bioactivities, and the result may root in the cooperation of multiple substances. So far, except for polyphenols, different kinds of bioactive constituents were founded in Hawk tea, such as organic acids, essential oils, saponins, and polysaccharides, some of them have been proved to be effective on α-glucosidase inhibition, and their content differences in different months may affect the correlation of enzymatic inhibition and TPC. As to these compounds, we are planning to research them in our next steps and hope to comprehensively take more bioactive chemicals into consideration.
Point 4: As a whole, I think this manuscript is topic is unclear. For example, Figure 2 and Figure 5 are far from topic of this manuscript title. Result section contained discussion, and discussion section contained result. It should be re consisted the sentence.
Response 4: In this work, the quantitative analysis of quercitrin and astragalin were processed by calculating their peak sum. So, the origin intention of Figure 2 and Figure 5 is to explain and demonstrate our quantitation. We reconsidered it according to your question and eliminate it now, for it was not highly relevant to our topic.
Point 5: Page 6 line 139, November →December?
Response 4: It should be December. Now we have corrected it, as shown in page 10, line 158.
Point 6: Page8 line183 to 190, 202 to 206, 212 to 216 This sentence is discussion. It is not results. It should be move to discussion section.
Response 6: The sentences mentioned above has been moved and modified in our revised manuscript. As shown below:
Lines 183 to 190→270 to 277,
Lines 202 to 206→286 to 290
and the sentences from lines 212 to 216 have been removed.
Point 7: P9 line237 Litsea coreane H →Scientific name should be written by italic.
Response 7: In the revised manuscript, all of the scientific name writings have been corrected to italic, such as the sentences in lines 33, 250, 279.
Point 8: P12 line 321 It should be described HPLC system name, flow speed, injection volume etc.
Response 8: The HPLC condition is put in more detail in line 359to 365, and also shown below:
Flavonoids were tested by HPLC (LC-20AD, Shimadzu, Kyoto, Japan) with a photo-diode array detector (DAD; Shimadzu, Kyoto, Japan). A column (Hypersil ODS2 5 μm, 4.6 mm × 250 mm i.d., SinoChrom, Dalian, China) with the mobile phase, aqueous 0.1% formic acid solution (A) and acetonitrile (B) and a flow rate of 1 mL/min with a 10 μL injection volume were used, and the column oven temperature was set to 30 °C. For better isolation, the gradient program was set as follows: 0–38 min, 11–20% B; 38–66 min, 20–40% B; 66–66.01 min, 40–11% B; and 66.01–70 min, 11% B. The detecting wavelength was set to 350 nm.
Round 2
Reviewer 1 Report
This version has been substantially improved and I recommend its publicaltion in Molecules after typos are corrected.
Reviewer 2 Report
The manuscript has been revised well.
Please check below.
Table 5. orlistat lipase IC50→0.0006
Page8 line222→IC50(0.006 mg/mL)